# An integrative examination of psychological distress and its nutritional and visual correlates among young adults in Ghana

Isaiah Osei Duah Junior[1,*,¤a], Wendy Ofori Asare[1], Elizabeth J. Johnson[2], Charllote Boateng[3,¤b], Hubert Osei Acheampong[1,¤c], David Ben Kumah[1], Kwadwo Owusu Akuffo[1]

**1** Department of Optometry and Visual Science, College of Science, Kwame Nkrumah University of Science and Technology, Kumasi, Ghana, **2** Friedman School of Nutrition Science and Policy, Tufts University, Boston, Massachusetts, United States of America, **3** School of Public Health, College of Health and Allied Sciences, Kwame Nkrumah University of Science and Technology, Kumasi, Ghana

¤a Current address: Department of Biological Sciences, College of Science, Purdue University, West Lafayette, Indiana, United States of America
¤b Current address: School of Public Health, University of Memphis, Memphis, Tennessee, United States of America
¤c Current address: Graduate Program in Anatomy and Cell Biology, School of Medicine, Wayne State University, Detroit, Michigan, United States of America
* oseiduahisaiah@gmail.com

## Abstract

Psychological distress is a key precursor to suicidal ideation during emerging adulthood, a period marked by increasing independence and responsibility. Despite this vulnerability, the prevalence and determinants of psychological distress among young adults in Ghana remain poorly understood. This study examined psychological distress and its lifestyle, dietary, visual, and cognitive correlates in 301 participants. Data on biographical, health, and physical activity factors were collected via structured questionnaires. Visual function was assessed objectively using LogMAR ETDRS charts and Pelli-Robson contrast sensitivity charts, and subjectively via the NEI-VFQ-25. Macular pigment optical density (MPOD) and dietary carotenoid intake were also measured. Cognitive function was assessed with a neuropsychological test battery, and psychological distress with the Kessler K10 scale. Ordinal logistic regression indicated that 23.3% of participants experienced psychological distress (14.6% mild, 3.0% moderate, 5.6% severe). Multivariate analyses showed that occasional alcohol consumption (AOR = 4.10, 95% CI: 1.63–10.12, p = 0.003) and longer swimming duration (AOR = 1.04, 95% CI: 1.01–1.09, p = 0.037) increased the odds of higher psychological distress severity, whereas higher VFQ–25 scores were protective (AOR = 0.95, 95% CI: 0.91–0.99, p = 0.008). Spending more than 50% of the day outdoors showed a marginal trend toward increased odds (AOR = 2.40, 95% CI: 0.97–5.81, p = 0.054). Other factors, including BMI, ethnicity, protective eyewear,

**Data availability statement:** All relevant data and materials supporting the conclusion of this article is/are available within the manuscript and its supporting information files.

**Funding:** The author(s) received no specific funding for this work.

**Competing interests:** The authors have declared that no competing interests exist.

**Abbreviations:** AOR, adjusted odds ratio; BMI, body mass index; BCVA, best-corrected visual acuity; FMI, fat mass index; CI, confidence interval; LM, lean mass; LMI, lean mass index; PSD, psychological distress; SMM, skeletal muscle mass; MPOD, macular pigment optical density; MMSE, Mini-Mental State Examination (MMSE); NEI-VFQ-25, National Eye Institute Visual Function Questionnaire-25 (NEI VFQ-25); SD, standard deviation; SMMI, skeletal muscle mass index; OR, odds ratio.

and dietary carotenoids, were not significant after adjustment. These findings suggest that specific lifestyle and visual function factors influence the severity of psychological distress in young adults, whereas demographic, anthropometric, and most dietary factors appear less influential on the severity of psychological distress. Collectively, integrating substance use counseling, visual health support, and lifestyle interventions may help reduce psychological distress and promote mental wellbeing during emerging adulthood.

## Introduction

Optimal mental health is essential for effective functioning, adaptive coping, and the maintenance of healthy social relationships. When mental health deteriorates, young adults may experience suicidal thoughts, representing a critical public health concern [1,2]. Emerging adulthood, generally encompassing the twenties, is a transitional period characterized by the assumption of personal responsibilities, independent decision-making, and the pursuit of financial and social autonomy [3]. This life stage is often marked by the formation of intimate relationships [4] alongside demanding academic or occupational commitments [5] which may limit opportunities for physical activity [6]. The pursuit of independence during this period may also promote sedentary behavior and engagement in risky lifestyle practices, including poor dietary habits, alcohol consumption, and tobacco use [7]. The combined influence of these factors can disrupt the balance between personal, academic, and social responsibilities, reducing resilience to stress and increasing the likelihood of psychological distress [8].

Psychological distress, defined as a mental health condition characterized by emotional suffering, anxiety, depressive symptoms, and feelings of nervousness, remains a well-established precursor to suicidal ideation [9]. In young adults, mild or unrecognized stress can develop into chronic psychological distress, which may contribute to cognitive difficulties and impaired functioning. Higher levels of psychological distress are associated with reduced cognitive performance, diminished attention and memory capacity, and accelerated cognitive decline [10]. Emerging evidence suggests that cognitive health may be enhanced through dietary intake of specific nutrients such as carotenoids, which could play a protective role against psychological distress [11–13].

Dietary carotenoids, particularly lutein and zeaxanthin, have been associated with improved cognitive performance throughout life [14]. These carotenoids are abundant in fruits, vegetables, sweet corn, and eggs, and are known to provide antioxidant, anti-inflammatory, neuroprotective, and light-filtering benefits that support both cognitive and visual health [14–17]. Despite these benefits, studies indicate that young adults often consume insufficient amounts of fruits and vegetables, resulting in low dietary intake of lutein and zeaxanthin [18–20]. Low levels of these carotenoids have been linked to cognitive decline and increased psychological stress, suggesting that dietary insufficiency may contribute to higher levels of psychological distress in this population [12].

Visual function is another factor closely linked to psychological well-being. Several studies have reported an inverse relationship between visual difficulties and psychological distress, with individuals experiencing visual problems often reporting higher levels of anxiety and emotional strain [21–26]. Emerging adulthood is associated with intense academic and occupational demands that place substantial stress on the visual system, increasing the risk of symptoms such as eye strain, headaches, and refractive errors [27,28]. Unrecognized or untreated visual impairments can exacerbate psychological distress, highlighting the importance of early detection and correction [26,27,29] and induced refractive errors [28]. Unrecognized and untreated visual issues can contribute to psychological distress in this at-risk population [26,29].

High-risk behaviors, particularly alcohol use, are prevalent among young adults [30–32], often driven by social pressure, experimentation, stress coping, and the need for peer acceptance [33–35]. Consequentially, alcohol consumption, especially during social events or episodic drinking, can impair memory, judgment, attention, and coordination, thereby increasing susceptibility to psychological distress [36]. Together, the interaction of impaired cognitive function, visual strain, and high-risk behaviors emphasizes the complex and multifactorial nature of psychological distress during emerging adulthood.

Psychological distress exists along a continuum from mild to severe. Early symptoms are frequently unrecognized or untreated, potentially progressing to severe mental health outcomes such as suicidal ideation and attempts. Young adults are particularly vulnerable due to the transitional nature of this developmental stage, which encompasses academic, social, and occupational challenges. Despite this vulnerability, the prevalence, burden, and underlying factors contributing to psychological distress among young adults in Ghana remain largely unexplored. Understanding these determinants is critical for informing targeted interventions and public health strategies aimed at promoting mental well-being.

This study therefore aimed to comprehensively assess psychological distress among young adults in Ghana, with a focus on nutritional and visual correlates. A multimodal approach was employed, combining self-reported questionnaires with clinician-administered assessments. The significance of this research lies in its potential to address critical gaps in mental health knowledge in Ghana, providing evidence-based insights into modifiable lifestyle factors, informing the design of early detection programs, and guiding policy initiatives to reduce the burden of psychological distress and enhance the overall quality of life among young adults.

## Materials and methods

### Study design, population, and area

This study utilized data collected between July 12 to August 2, 2021, as part of a larger research project evaluating macular pigment optical density (MPOD) in a cohort of healthy young adults in Ghana [20]. In brief, systematic random sampling combined with proportionate to size was used to enroll 301 healthy volunteers (defined as subjects with no ocular lesions and no neurological diseases who could tolerate the MPOD test). A hierarchical methodology was used, with biographical data, health status, lifestyle, physical activity, diet, anthropometry, visual function, neuropsychological performance and psychological distress collected either by structured questionnaire or standardized clinical assessment, as appropriate, and where necessary, by an optometric resident and well-trained clinical research assistants [20].

### Measures

Biographical variables in our analysis include age, sex, ethnicity; health status variables include systolic and diastolic blood pressure and medication use; lifestyle variables include smoking history, exposure to passive smoking, alcohol consumption, light exposure, protective eyewear and physical activity measures; type and duration of physical activity.

### Anthropometric measures

Bioelectric impedance analysis using the Seca mBCA525 portable device was used to assess fat mass index (FMI) and skeletal muscle mass index (SMMI), together with estimation of body weight and height to calculate body mass index

(BMI) [37]. BMI categorized participants into underweight (<18.5), standard and/or healthy (18.5–24.9), overweight (25.0–29.9), and obese (≥30) [38].

### Dietary measures: lutein and zeaxanthin intake

Twenty-four-hour dietary recall of a three-day dietary pattern (two weekdays and one weekend) guided by food modules and household handles assessed as previously described to investigate carotenoid intake [20,39]. Lutein and zeaxanthin intakes were estimated with a validated dietary questionnaire [20].

### Visual function measures

Visual function data were collected by optometric residents. Briefly, monocular visual acuity was measured using the logarithm of the minimum angle of resolution (LogMAR) of the Early Treatment Diabetic Retinopathy Study (ETDRS) at four meters and contrast sensitivity function was assessed using the Pelli-Robson chart at one metre [20]. In vivo macular pigment optical density, a surrogate measure of retinal and brain carotenoids, was measured at half and one degree of retinal eccentricity using adapted heterochromatic flicker photometry [20]. The MPOD utilizes iso-illuminance matching of the peripheral, non-absorbed green flicker and the central, maximally absorbed, blue, non-flicker circular stimulus (log $I_{central}$/$I_{peripheral}$) [40]. MPOD data were quantified from the eyes with the best corrected visual acuity (BCVA) and/or the dominant eyes using the Miles technique for equal monocular BCVA [41]. Further, subjective visual function, which measures the impact of visual impairment on various aspects of a person's life, including daily activities, social functioning and emotional well-being, was assessed using the National Eye Institute Visual Function Questionnaire-25 (NEI VFQ-25). The NEI VFQ-25 scale ranges from 0 to 100, with higher scores indicating better visual functioning [42].

### Neuropsychological function measures

Global cognition (attention, orientation, memory, and spatial recognition) were assessed with a 30-point dementia and/or Alzheimer's disease screening tool, the Mini-Mental State Examination (MMSE) [20,43], phonetic verbal fluency was investigated with the "FAS Test" as a function of language, memory, and executive functions; and Animal fluency test was used to interrogate the semantic memory and verbal fluency [20,44].

### Psychological distress measures

The primary dependent variable was psychological distress, measured by the Kessler-10 Psychological Distress Scale (K-10) [45]. The scale consists of 10 items (e.g., 'In the past 4 weeks, how often have you felt so nervous that nothing could calm you down?'), with each item having a five-point response scale: 'always' (5), 'most of the time' (5), 'most of the time' (4), 'sometimes' (3), 'some of the time' (2), and 'never'. The scale ranges from 10 to 50, with a score < 20 indicating low/minimal and/or no distress, 20−24 indicating mild distress, 25−29 indicating moderate distress, and a score ≥ 30 indicating severe distress [20]. The Kessler-10 scale is related to the Composite International Diagnostic Interview (CIDI), a standard tool for the assessment of mental disorders, and remains a robust tool for the assessment of psychological distress [45,46].

### Ethical consideration

The study was approved by the Institutional Review Board of the Kwame Nkrumah University of Science and Technology, Committee on Human Research Publication and Ethics at the School of Medicine and Dentistry (reference number: CHRPE/AP/198/21). Written informed consent was obtained voluntarily from all participants before enrolment, and all procedures used in the study conformed to the tenets of the Declaration of Helsinki.

## Data analysis

Data were analyzed using the Statistical Product and Service Solution (SPSS) version 25, compatible with Windows 10, and RStudio (version 4.3.0) for additional statistical analyses and data visualization. The Kolmogorov–Smirnov test was employed to assess the normality of continuous variables. Where appropriate, group differences were analyzed using independent one-way ANOVA with F-statistic and Eta-squared ($\eta^2$) effect size for normally distributed continuous variables, Kruskal–Wallis one-way ANOVA with Kruskal–Wallis statistic (H), epsilon-squared ($\varepsilon^2$) effect size, and mean rank for non-normally distributed continuous variables, or chi-squared ($\chi^2$) analysis with Pearson chi-square statistic and Cramer's V (V) effect size for categorical variables. Statistical significance was determined at $p \leq 0.05$. To examine the association between each explanatory variable and the severity of psychological distress, an ordinal logistic regression analysis was performed. Initially, bivariate analyses were conducted using the Wald test at a significance threshold of $p \leq 0.20$ to identify candidate variables. Those that met this criterion were subsequently included in the multivariate ordinal logistic regression model, with statistical significance set at $p \leq 0.05$. Adjusted odds ratios (AOR) with 95% confidence intervals (CI) were computed to quantify the strength and direction of associations. The outcome variable, psychological distress severity, was categorized according to the Kessler-10 (K10) scale as follows: no distress (K10 < 20), mild distress (K10 = 20–24), moderate distress (K10 = 25–29), and severe distress (K10 ≥ 30). Figures were generated using Graph Pad Prism version 10.6.1 (compatible with macOS), and a concept map illustrating variable interactions was created using BioRender. For comparative analyses, psychological distress (PSD) was also assessed on a dichotomous scale (distressed [PSD] vs. non-distressed [No PSD]) [24,47].

## Results

### Psychological distress is linked to ethnicity, alcohol use, and eyewear factors but not to physiological or physical activity measures

Psychological distress was unevenly distributed in the sample, with most participants reporting no distress (76.7%), followed by mild (14.6%), severe (5.6%), and moderate distress (3.0%). Biographical characteristics such as age and sex showed no notable differences in distress levels, whereas ethnicity demonstrated a small but significant distribution effect (p = 0.013). Health measures, including blood pressure and medication use, were evenly distributed across distress categories (p > 0.05). Lifestyle factors largely showed uniform distributions, though alcohol consumption exhibited a significant difference (p = 0.038), indicating higher distress among some drinkers. Use of eyewear was not significantly associated with distress levels (p = 0.176), although the form of eyewear showed a significant difference (p = 0.016), while the purpose and frequency of use did not significantly affect distress distribution (p > 0.05). Physical activity patterns, including type, frequency, and total activity scores, were not significantly associated with differences in distress distribution (p > 0.05) see Table 1.

### Psychological distress is linked to lower carotenoid intake and poorer subjective visual function but not objective physiological or cognitive measures

Table 2 presents the relationship between psychological distress and a range of dietary, anthropometric, visual, and cognitive measures among participants (see Table 2). Psychological distress was significantly associated with lower dietary intake of lutein, zeaxanthin, and their combined intake (all p < 0.05; see Fig 1).
    Participants with psychological distress had substantially lower consumption of these carotenoids compared with those without distress. However, when levels of distress severity (mild, moderate, severe) were examined, this difference was annulled, suggesting that the relationship between carotenoid intake and distress was more pronounced in the presence versus absence of distress, but did not vary significantly across the different severity levels (all p > 0.05; see Table 2). Vitamin or supplement use was not significantly related to distress (p = 0.338), indicating that supplement habits were not associated

**Table 1. Biographical, health, lifestyle, and physical activity correlates of psychological distress among young adults from Ghana.**

| Variable | Psychological distress % (n) | | | | Test-statistic | Effect size | p-value |
|---|---|---|---|---|---|---|---|
| **Level of psychological distress** | **No distress** | **Mild distress** | **Moderate distress** | **Severe distress** | | | |
| Kessler scale | (K10 ≤ 19) | (K10 20–24) | (K10 25–29) | (K10 ≥ 30) | | | |
| Prevalence of PSD | 76. 7 (231) | 14.6 (44) | 3.0 (9) | 5.6 (17) | | | |
| **Biographical measures** | | | | | | | |
| Age, years (mean ± SE) | 21.16 ± 0.13 | 20.98 ± 0.31 | 20.56 ± 0.58 | 21.06 ± 0.46 | F = 0.378 | $\eta^2$ = 0.004 | p = 0.769 |
| *Sex* | | | | | | | |
| Male | 41.2 (124) | 6.6 (20) | 1.7 (5) | 2.3 (7) | $\chi^2$ = 1.858 | V = 0.079 | p = 0.603 |
| Female | 35.5 (107) | 8.0 (24) | 1.3 (4) | 3.3 (10) | | | |
| *Ethnicity* | | | | | | | |
| Akan | 61.1 (184) | 10.6 (32) | 2.7 (8) | 4.3 (13) | $\chi^2$ = 29.619 | V = 0.181 | p = 0.013 |
| Ga | 2.7 (8) | 0.7 (2) | 0.0 (0) | 0.0 (0) | | | |
| Ewe | 7.0 (21) | 0.7 (2) | 0.3 (1) | 0.3 (1) | | | |
| Northerners | 4.0 (12) | 1.7 (5) | 0.0 (0) | 0.0 (0) | | | |
| Guan | 1.0 (3) | 0.7 (2) | 0.0 (0) | 0.0 (0) | | | |
| Others | 1.0 (3) | 0.3 (1) | 0.0 (0) | 1.0 (3) | | | |
| **Health status measures** | | | | | | | |
| *Blood pressure, mm Hg (mean ± SE)* | | | | | | | |
| Diastolic | 70.33 ± 0.58 | 69.48 ± 1.26 | 64.89 ± 3.09 | 70.29 ± 2.07 | F = 1.206 | $\eta^2$ = 0.012 | p = 0.308 |
| Systolic | 107.82 ± 0.90 | 106.11 ± 2.07 | 100.67 ± 2.91 | 106.76 ± 2.78 | F = 0.692 | $\eta^2$ = 0.010 | p = 0.406 |
| *Medication Use* | | | | | | | |
| No | 72.1 (217) | 13.6 (41) | 3.0 (9) | 4.7 (14) | $\chi^2$ = 4.087 | V = 0.117 | p = 0.252 |
| Yes | 4.7 (14) | 1.0 (3) | 0.0 (0) | 1.0 (3) | | | |
| **Lifestyle measures** | | | | | | | |
| *Secondhand smoke* | | | | | | | |
| Not Exposed | 71.1 (214) | 13.0 (39) | 2.7 (8) | 4.7 (14) | $\chi^2$ = 2.711 | V = 0.095 | p = 0.438 |
| Exposed | 5.6 (17) | 1.7 (5) | 0.3 (1) | 1.0 (3) | | | |
| *Alcohol consumption* | | | | | | | |
| None (never drink) | 69.8 (210) | 12.6 (38) | 2.7 (8) | 3.7 (11) | $\chi^2$ = 13.350 | V = 0.149 | p = 0.038 |
| Special occasion | 5.3 (16) | 2.0 (6) | 0.3 (1) | 1.7 (5) | | | |
| 1–2 times a month | 1.7 (5) | 0.0 (0) | 0.0 (0) | 0.3 (1) | | | |
| *Volume of alcohol consumption* | | | | | | | |
| 1 units per week | 76.5 (13) | 5.9 (1) | 0.0 (0) | 11.8 (2) | $\chi^2$ = 4.958 | V = 0.540 | p = 0.084 |
| 6–10 units per week | 0.0 (0) | 0.0 (0.0) | 0.0 (0) | 5.9 (1) | | | |
| *Light exposure* | | | | | | | |
| Little (<50% of the day outside) | 34.2 (103) | 4.3 (13) | 0.7 (2) | 3.0 (9) | $\chi^2$ = 6.006 | V = 0.141 | p = 0.422 |
| Some (50% outside and 50% inside) | 32.9 (99) | 7.6 (23) | 1.7 (5) | 2.0 (6) | | | |
| Most (Spends >50% of the time outside) | 9.6 (29) | 2.7 (8) | 0.7 (2) | 0.7 (2) | | | |
| *Use of eye wear* | | | | | | | |
| Yes | 26.2 (79) | 7.0 (21) | 1.7 (5) | 2.7 (8) | $\chi^2$ = 4.950 | V = 0.128 | p = 0.176 |
| No | 50.5 (152) | 7.6 (23) | 1.3 (4) | 3.0 (9) | | | |
| *Form of eye wear* | | | | | | | |
| None | 57.5 (173) | 8.0 (24) | 1.3 (4) | 3.7 (11) | $\chi^2$ = 15.631 | V = 0.161 | p = 0.016 |
| Spectacles | 19.3 (58) | 6.3 (19) | 1.7 (5) | 2.0 (6) | | | |
| Contact lens | 0.0 (0) | 0.3 (1) | 0.0 (0) | 0.0 (0) | | | |

*(Continued)*

**Table 1.** (Continued)

| Variable | Psychological distress % (n) | | | | Test-statistic | Effect size | p-value |
|---|---|---|---|---|---|---|---|
| Level of psychological distress | No distress | Mild distress | Moderate distress | Severe distress | | | |
| *Purpose of eye wear* | | | | | | | |
| Distant activity | 37.1 (33) | 10.1 (9) | 2.2 (2) | 3.4 (3) | $\chi^2 = 15.631$ | V = 0.117 | p = 0.873 |
| Near activity | 7.9 (7) | 1.1 (1) | 1.1 (1) | 1.1 (1) | | | |
| Both | 21.3 (19) | 10.1 (9) | 2.2 (2) | 2.2 (2) | | | |
| *Frequency of use of eyewear* | | | | | | | |
| Occasionally | 10.1 (9) | 3.4 (3) | 1.1 (1) | 0.0 (0) | $\chi^2 = 6.612$ | V = 0.193 | p = 0.358 |
| Part-time | 24.7 (22) | 11.2 (10) | 0.0 (0) | 2.2 (2) | | | |
| Full-time | 31.5 (28) | 6.7 (6) | 4.5 (4) | 4.5 (4) | | | |
| **Physical activity measures** | | | | | | | |
| *Type of physical activity* | | | | | | | |
| Walking, yes | 76.7 (231) | 14.6 (44) | 3.0 (9) | 5.6 (17) | nd | nd | nd |
| Running, yes | 10.3 (31) | 1.3 (4) | 0.3 (1) | 0.0 (0) | $\chi^2 = 3.127$ | V = 0.102 | p = 0.372 |
| Swimming, yes | 1.7 (5) | 1.0 (3) | 0.0 (0) | 0.3 (1) | $\chi^2 = 3.533$ | V = 0.108 | p = 0.316 |
| Cycling, yes | 1.7 (5) | 0.0 (0) | 0.0 (0) | 0.3 (1) | $\chi^2 = 2.429$ | V = 0.090 | p = 0.488 |
| Gym-based workout, yes | 7.3 (22) | 2.7 (8) | 0.0 (0) | 0.0 (0) | $\chi^2 = 6.238$ | V = 0.144 | p = 0.101 |
| Team-based sports, yes | 5.6 (17) | 0.3 (1) | 0.0 (0) | 0.3 (1) | $\chi^2 = 2.254$ | V = 0.087 | p = 0.521 |
| *Duration of physical activity in mins* | | | | | | | |
| Walking | | | | | | | |
| Running | 11.30 ± 3.53 | 2.23 ± 1.33 | 1.56 ± 1.56 | 0.0 ± 0.0 | F = 0.760 | $\eta^2 = 0.008$ | p = 0.517 |
| Swimming‡ | 149.71 | 156.9 | 146.5 | 155.59 | H = 3.719 | $\varepsilon^2 = 0.002$ | p = 0.293 |
| Cycling | 1.21 ± 0.58 | 0.0 ± 0.0 | 0.0 ± 0.0 | 0.29 ± 0.29 | F = 0.399 | $\eta^2 = 0.004$ | p = 0.754 |
| Gym-based workout‡ | 149.79 | 163.28 | 135.5 | 143.82 | H = 4.751 | $\varepsilon^2 = 0.006$ | p = 0.191 |
| Team-based sports‡ | 10 | 2 | | 8 | H = 4.151 | $\varepsilon^2 = 0.077$ | p = 0.125 |
| Total physical activity scores | 380.42 ± 25.00 | 416.66 ± 75.04 | 473.11 ± 148.18 | 380.18 ± 137.74 | F = 0.226 | $\eta^2 = 0.002$ | p = 0.878 |

**Note:** *n*, frequency (may not equal 301 for all variables); %, percentage frequency; SE, standard error. Normality of the data was assessed using the Kolmogorov–Smirnov test. Where appropriate, group differences were analyzed using independent one-way ANOVA with *F*-statistic and Eta-squared ($\eta^2$) effect size, Kruskal–Wallis one-way ANOVA with Kruskal–Wallis statistic (*H*) and epsilon-squared ($\varepsilon^2$) effect size with ‡ mean rank, or chi-squared analysis with Pearson chi-square statistic ($\chi^2$) and Cramer's *V* (V) effect size. Statistical significance was set at $p \leq 0.05$.

with psychological health. Body composition indices, including fat mass, visceral fat, lean mass, muscle mass, and BMI, showed no significant variation among the distress groups (all p > 0.05). Participants with severe psychological distress had somewhat higher BMI and body fat values on average, but these differences were not statistically significant (p > 0.05). When BMI was categorized into healthy, underweight, overweight, and obese, no significant differences were found (p = 0.125), further suggesting that body composition did not play a major role in explaining psychological distress. Visual performance measures, including best-corrected visual acuity, contrast sensitivity, and macular pigment optical density (measured at 0.5° and 1°), showed no significant differences across distress levels (all p > 0.05). However, subjective visual function (VFQ-25 scores) differed significantly between individuals with and without psychological distress (p = 0.001; see **Fig 2**), and this difference persisted across levels of distress severity (p = 0.002; see Table 2). Participants with low distress reported the highest perceived visual function, whereas those with moderate distress reported the lowest, suggesting that perceived rather than objective visual quality was more closely linked to emotional well-being. Neuropsychological measures showed a similar pattern. Global cognitive function, assessed by the Mini-Mental State Examination (MMSE), approached significance (p = 0.056), with participants experiencing mild distress scoring slightly higher than others. Verbal fluency measures (F-A-S and Animal Fluency) did not differ significantly (p > 0.05), indicating stable cognitive performance across distress levels (see **Table 2**).

**Table 2. Psychological distress in relation to dietary intake, body composition, visual performance, and cognitive measures among young adults from Ghana.**

| Variable | Psychological distress | | | | Test-statistic | Effect size | p-value |
|---|---|---|---|---|---|---|---|
| Level of psychological distress | No distress | Mild distress | Moderate distress | Severe distress | | | |
| | Mean ± SE | Mean ± SE | Mean ± SE | Mean ± SE | | | |
| **Dietary measures** | | | | | | | |
| Lutein (L), *mg/d* | 4.47 ± 0.74 | 3.02 ± 1.32 | 0.66 ± 0.27 | 3.01 ± 1.81 | F = 0.629 | $\eta^2$ = 0.006 | p = 0.597 |
| Zeaxanthin (Z), *mg/d* | 0.37 ± 0.17 | 0.20 ± 0.04 | 0.11 ± 0.03 | 0.13 ± 0.04 | F = 0.147 | $\eta^2$ = 0.001 | p = 0.932 |
| L+Z, *mg/d* | 4.84 ± 0.82 | 3.22 ± 1.32 | 0.92 ± 0.31 | 3.14 ± 1.80 | F = 0.600 | $\eta^2$ = 0.001 | p = 0.615 |
| *Regular vitamins or supplement* | | | | | | | |
| Vitamins or supplement use | 8.6 (26) | 2.7 (8) | 0.3 (1) | 1.3 (4) | $\chi^2$ = 3.370 | V = 0.106 | p = 0.338 |
| No vitamins or supplement | 68.1 (205) | 12.0 (36) | 2.7 (8) | 4.3 (13) | | | |
| **Anthropometric measures** | | | | | | | |
| Fat Mass or Body Fat (FM) | 26.25 ± 1.64 | 27.92 ± 1.76 | 22.31 ± 3.33 | 30.25 ± 3.84 | F = 0.324 | $\eta^2$ = 0.003 | p = 0.808 |
| Fat Mass Index in kg/m$^2$ (FMI) | 9.61 ± 0.68 | 10.31 ± 0.72 | 7.75 ± 1.31 | 11.04 ± 1.44 | F = 0.308 | $\eta^2$ = 0.003 | p = 0.820 |
| Visceral Fat (FT) | 4.38 ± 0.15 | 4.91 ± 0.15 | 4.33 ± 0.65 | 4.71 ± 0.57 | F = 0.657 | $\eta^2$ = 0.007 | p = 0.579 |
| Lean Mass (LM) | 37.70 ± 1.74 | 38.43 ± 2.21 | 43.94 ± 5.02 | 42.13 ± 3.78 | F = 0.348 | $\eta^2$ = 0.004 | p = 0.790 |
| Lean Mass Index in kg/m$^2$ (LMI) | 13.04 ± 0.66 | 13.51 ± 0.63 | 14.61 ± 1.39 | 15.03 ± 1.26 | F = 0.341 | $\eta^2$ = 0.003 | p = 0.796 |
| Muscle Mass (MM)‡ | 153.07 | 132.44 | 148.39 | 146.38 | H = 2.139 | $\varepsilon^2$ = 0.002 | p = 0.544 |
| Waist circumference‡ | 147.67 | 152.36 | 158.5 | 179.76 | H = 1.739 | $\varepsilon^2$ = 0.004 | p = 0.628 |
| *BMI in kg/m$^2$* | 23.77 ± 0.86 | 24.50 ± 0.94 | 22.41 ± 0.80 | 26.0 ± 2.03 | F = 0.280 | $\eta^2$ = 0.003 | p = 0.840 |
| *BMI in kg/m$^{2(Cat)}$* | | | | | | | |
| Healthy | 50.0 (149) | 8.1 (24) | 2.7 (8) | 3.4 (10) | $\chi^2$ = 13.924 | V = 0.125 | p = 0.125 |
| Underweight | 6.7 (20) | 1.3 (4) | 0.0 (0.0) | 0.0 (0.0) | | | |
| Overweight | 16.4 (49) | 3.0 (9) | 0.3 (1) | 1.7 (5) | | | |
| Obese | 3.4 (10) | 2.3 (7) | 0.0 (0.0) | 0.7 (2) | | | |
| **Visual performance measures** | | | | | | | |
| *Best corrected visual acuity* | | | | | | | |
| Oculus dexter | −0.20 ± 0.01 | −0.21 ± 0.01 | −0.22 ± 0.03 | −0.22 ± 0.03 | F = 0.432 | $\eta^2$ = 0.004 | p = 0.730 |
| Oculus sinister | −0.20 ± 0.01 | −0.21 ± 0.01 | −0.21 ± 0.01 | −0.19 ± 0.02 | F = 0.112 | $\eta^2$ = 0.001 | p = 0.593 |
| *Contrast sensitivity function* | | | | | | | |
| Oculus dexter | 1.61 ± 0.002 | 1.61 ± 0.004 | 1.62 ± 0.008 | 1.62 ± 0.006 | F = 0.337 | $\eta^2$ = 0.003 | p = 0.799 |
| Oculus sinister | 1.61 ± 0.004 | 1.61 ± 0.004 | 1.62 ± 0.008 | 1.62 ± 0.007 | F = 0.075 | $\eta^2$ = 0.001 | p = 0.973 |
| *Macular pigment optical density* | | | | | | | |
| MPOD at 1° | 0.35 ± 0.01 | 0.32 ± 0.03 | 0.33 ± 0.03 | 0.32 ± 0.03 | F = 0.516 | $\eta^2$ = 0.005 | p = 0.672 |
| MPOD at 0.5° | 0.37 ± 0.01 | 0.35 ± 0.03 | 0.38 ± 0.03 | 0.33 ± 0.04 | F = 0.576 | $\eta^2$ = 0.006 | p = 0.631 |
| *Subjective visual function* | | | | | | | |
| NEI- VFQ-25‡ | 160.6 | 130.56 | 85.2 | 107.97 | H = 14.481 | $\varepsilon^2$ = 0.039 | p = 0.002 |
| **Neuropsychological function** | | | | | | | |
| MMSE‡ | 147.55 | 180.98 | 128.83 | 132.09 | H = 7.576 | $\varepsilon^2$ = 0.015 | p = 0.056 |
| F-A-S | 43.59 ± 0.73 | 43.64 ± 1.89 | 40.33 ± 2.92 | 40.53 ± 2.28 | F = 0.625 | $\eta^2$ = 0.006 | p = 0.599 |
| Animal Fluency | 16.18 ± 0.25 | 17.48 ± 0.60 | 16.44 ± 1.06 | 15.82 ± 1.12 | F = 1.150 | $\eta^2$ = 0.015 | p = 0.212 |

**Note:** *n*, frequency (may not equal 301 for all variables); %, percentage frequency; SE, standard error. Normality of the data was assessed using the Kolmogorov–Smirnov test. Where appropriate, group differences were analyzed using independent one-way ANOVA with *F*-statistic and Eta-squared ($\eta^2$) effect size, Kruskal–Wallis one-way ANOVA with Kruskal–Wallis statistic (*H*) and epsilon-squared ($\varepsilon^2$) effect size with ‡ mean rank, or chi-squared analysis with Pearson chi-square statistic ($\chi^2$) and Cramer's *V* (V) effect size. Statistical significance was set at $p \le 0.05$. MMSE, Mini-Mental State Examination; F-A-S, phonemic verbal fluency test; NEI-VFQ-25, National Eye Institute Visual Function Questionnaire–25; MPOD, macular pigment optical density.

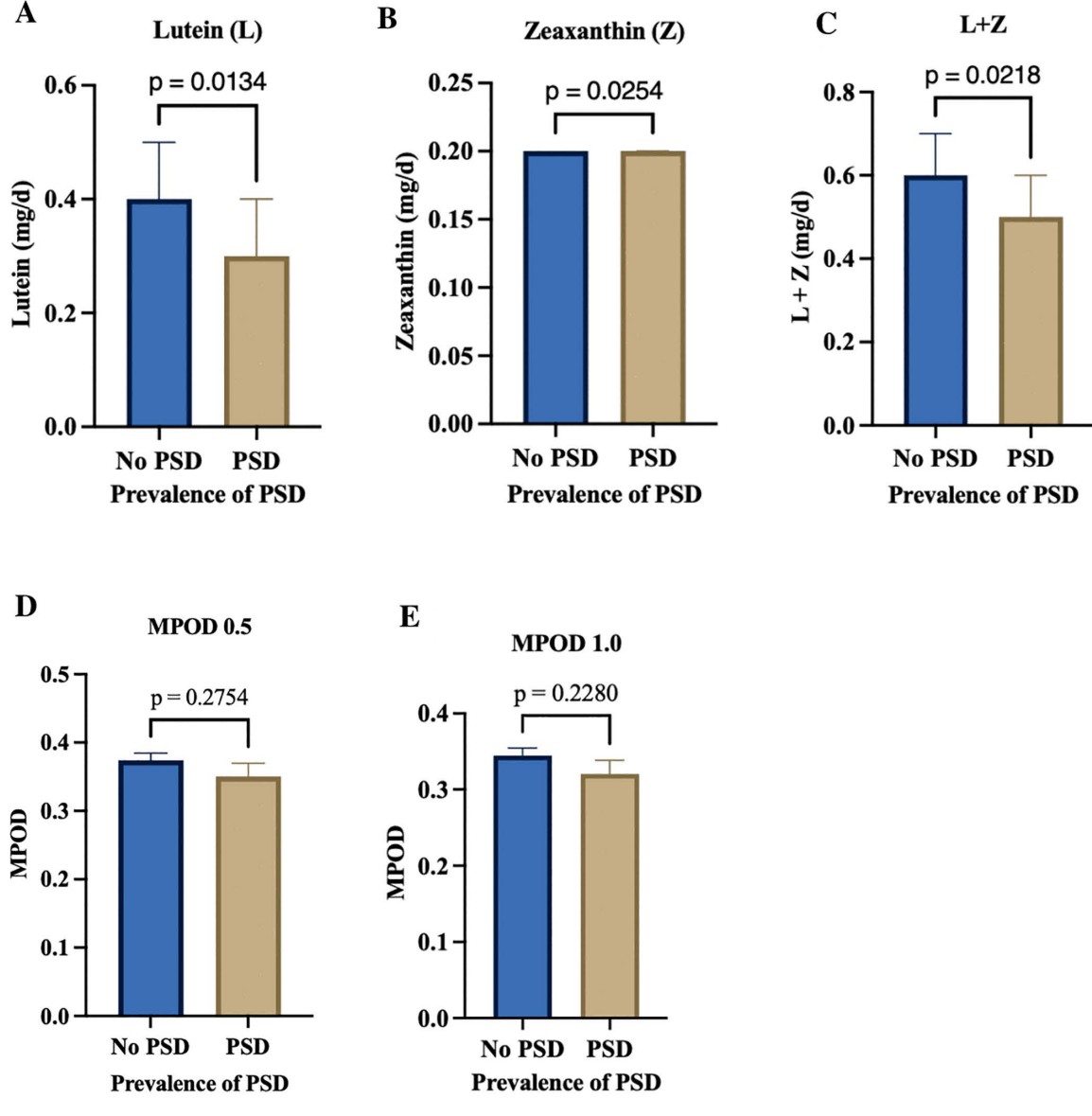

**Fig 1. Dietary and macular carotenoids by PSD status.** Mann–Whitney U tests indicated significant group differences for dietary lutein (p = 0.0134), zeaxanthin (p = 0.0254), and combined lutein + zeaxanthin (p = 0.0218) see top panel (A-C), but not for macular carotenoids where unpaired t-tests showed no significant differences in MPOD at 0.5° (p = 0.2754) or 1.0° of retinal eccentricity (p = 0.2280) see bottom panel (D-E).

### Factors influencing the severity of psychological distress

The results of the ordinal logistic regression analyses suggest that several lifestyles, dietary, and visual performance factors are associated with the severity of psychological distress, while many biographical, anthropometric, and physical activity measures were not significant (see **Table 3**). In the bivariate analyses, alcohol consumption on occasion, use of protective eyewear, spectacles, swimming duration, and VFQ-25 scores were significantly associated with severity of psychological distress. In the multivariate model, occasional alcohol consumption (AOR = 4.10, 95% CI: 1.63–10.12, p = 0.003) was associated with a significant increase in the odds of higher severity of psychological distress. Spending more than

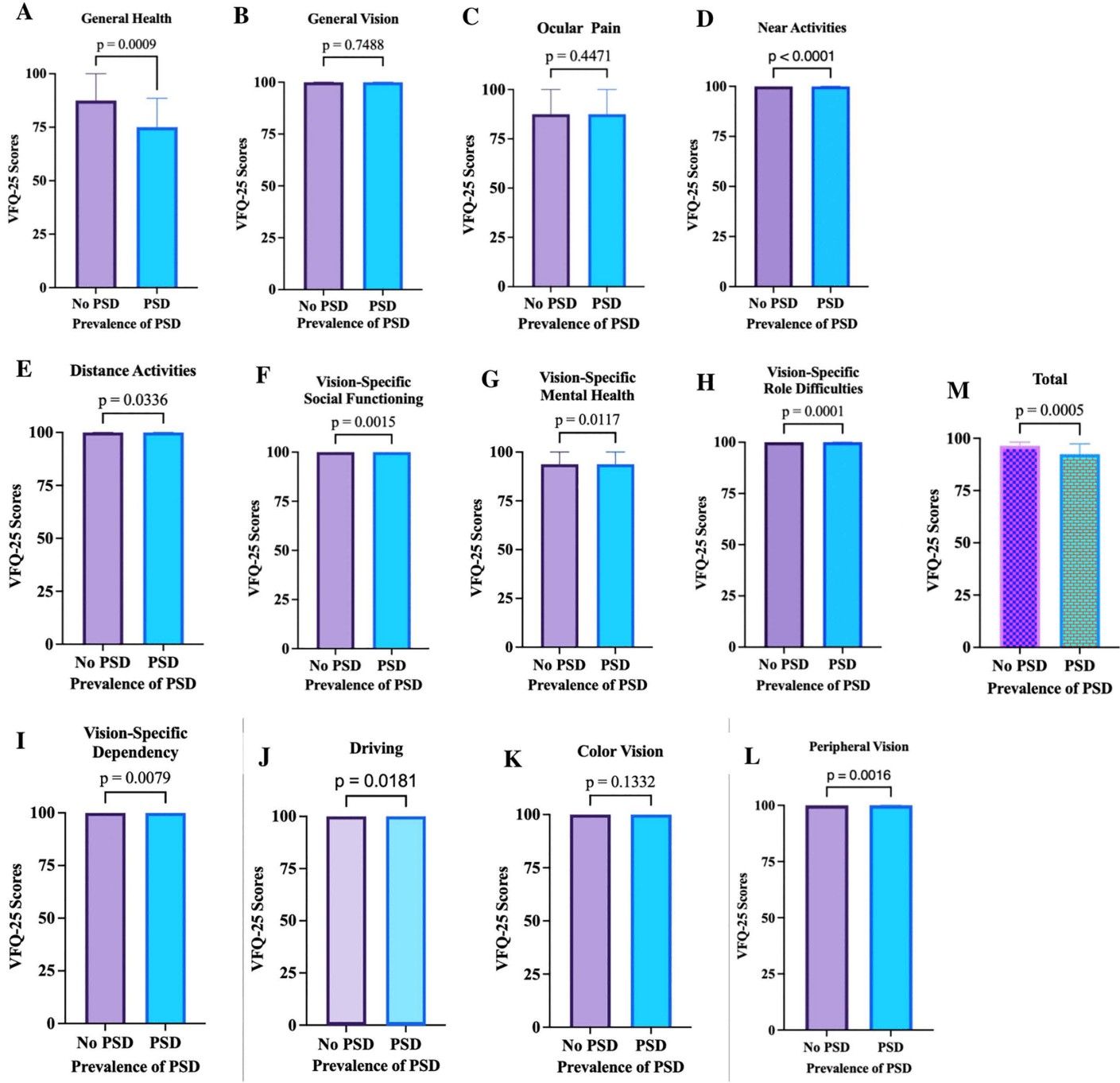

**Fig 2. Comparison of VFQ-25 Scores by PSD Status.** A Mann–Whitney U-test was conducted to compare individualized and composite scores of the short version of the VFQ-25 among participants with and without psychological distress (PSD) at a significance level of $p \le 0.05$. Participants with PSD reported significantly lower scores in several subscales, including (A) General Health ($U=6034$, $p=0.009$), (D) Near Activities ($U=6015$, $p<0.001$), (E) Distance Activities ($U=7013$, $p=0.0036$), (F) Vision-Specific Social Functioning ($U=6976$, $p=0.0015$), (G) Vision-Specific Mental Health ($U=6572$, $p=0.0117$), (H) Vision-Specific Role Difficulties ($U=6171$, $p=0.0001$), (I) Vision-Specific Dependency ($U=7238$, $p=0.0079$), (J) Driving ($U=6935$, $p=0.0181$), (L) Peripheral Vision ($U=6856$, $p=0.0016$), and (M) Total VFQ-25 score ($U=5866$, $p=0.0005$). Conversely, no statistically significant differences were observed for (B) General Vision ($U=7894$, $p=0.7488$), (C) Ocular Pain ($U=7611$, $p=0.4471$), or (K) Color Vision ($U=7723$, $p=0.1332$). Note. PSD dichotomized into present (mild, moderate, severe distress) and absent (no distress).

**Table 3. Bivariate and multiple ordinal regression analysis of factors associated to severity of psychological distress.**

| Variable | Bivariate ordinal logistic regression | | | Multivariate ordinal regression | | |
|---|---|---|---|---|---|---|
| | COR | 95% CI (Lower – Upper) | p-value | AOR | 95% CI (Lower – Upper) | p-value |
| **Biographical measures** | | | | | | |
| Age, years | 0.94 | 0.82 - 1.08 | 0.408 | | | |
| *Sex* | | | | | | |
| Male | Ref | | | | | |
| Female | 1.37 | 0.80 - 2.34 | 0.243 | | | |
| *Ethnicity* | | | | | | |
| Akan | Ref | | | | | |
| Ewe | 0.49 | 0.11 - 1.50 | 0.269 | | | |
| Ga | 1.23 | 0.27 - 4.25 | 0.758 | | | |
| Northerners | 1.24 | 0.39 - 3.36 | 0.69 | | | |
| Guan/Others | 3.79 | 1.17 - 11.71 | **0.022** | 3.57 | 0.69 - 16.94 | 0.115 |
| **Health status measures** | | | | | | |
| *Blood pressure, mm Hg* | | | | | | |
| Diastolic | 0.98 | 0.95-1.01 | 0.29 | | | |
| Systolic | 0.99 | 0.97- 1.01 | 0.219 | | | |
| *Medication Use* | | | | | | |
| Yes | 1.57 | 0.58-4.26 | 0.371 | | | |
| No | Ref | | | | | |
| **Lifestyle measures** | | | | | | |
| *Exposure to secondhand smoke* | | | | | | |
| Exposed to secondhand smoke | 1.91 | 0.82-4.45 | 0.133 | 1.79 | 0.66 - 4.52 | 0.234 |
| Not Exposed to secondhand smoking | Ref | | | | | |
| *Alcohol consumption* | | | | | | |
| No alcohol | Ref | | | | | |
| Alcohol on occasion | 2.99 | 1.33-6.50 | **0.007** | 4.10 | 1.63 - 10.12 | **0.003** |
| Alcohol use once or twice a month | 0.88 | 0.04-5.82 | 0.909 | | | |
| *Light exposure* | | | | | | |
| Spends <50% of the day outside | Ref | | | | | |
| 50% outside and 50% inside | 1.4 | 0.78-2.54 | 0.263 | | | |
| Spends >50% of the time outside | 1.67 | 0.74 - 3.66 | 0.206 | 2.40 | 0.97 - 5.81 | 0.054 |
| *Protective eyewear* | | | | | | |
| Yes | 1.8 | 1.05 - 3.09 | **0.032** | 1.11 | 0.50 - 2.39 | 0.797 |
| No | Ref | | | | | |
| *Form of eye wear* | | | | | | |
| None | | | | | | |
| Spectacles | 2.22 | 1.27 - 3.85 | **0.005** | 1.28 | 0.55 - 3.00 | 0.567 |
| Contact lenses | 7.99 | 0.29 - 222.97 | 0.164 | 2.56 | 0.07 - 98.02 | 0.582 |
| *Purpose of eye wear* | | | | | | |
| Distant | Ref | | | | | |
| Near | 1.15 | 0.22 - 4.78 | 0.855 | | | |
| Both | 1.52 | 0.60 - 3.82 | 0.375 | | | |
| *Frequency of use of eye wear* | | | | | | |
| Occasionally | Ref | | | | | |
| Part-time | 1.20 | 0.33 - 5.02 | 0.792 | | | |
| Full-time | 1.32 | 0.37 - 5.47 | 0.676 | | | |

*(Continued)*

**Table 3.** (Continued)

| Variable | Bivariate ordinal logistic regression | | | Multivariate ordinal regression | | |
|---|---|---|---|---|---|---|
| | COR | 95% CI (Lower – Upper) | p-value | AOR | 95% CI (Lower – Upper) | p-value |
| **Physical activity measures** | | | | | | |
| *Routine physical activity* | | | | | | |
| Running, yes | 0.48 | 0.18 - 1.28 | 0.143 | 1.51 | 0.23 - 8.62 | 0.649 |
| Swimming, yes | 2.44 | 0.68 - 8.73 | 0.17 | 0.01 | 0.00 - 0.55 | 0.085 |
| Cycling, yes | 0.78 | 0.09 - 6.98 | 0.821 | | | |
| Gym-based workout, yes | 1.06 | 0.46-2.47 | 0.886 | | | |
| *Duration of physical activity* | | | | | | |
| Walking | 1.00 | 1.00 - 1.00 | 0.526 | | | |
| Running | 0.98 | 0.95 - 1.01 | 0.109 | 0.97 | 0.91 - 1.00 | 0.142 |
| Swimming | 1.01 | 1.00 - 1.02 | **0.008** | 1.04 | 1.01 - 1.09 | **0.037** |
| Cycling | 0.95 | 0.83 - 1.09 | 0.463 | | | |
| Gym-based workout | 1.00 | 1.00 - 1.00 | 0.724 | | | |
| Team-based sports | 1.01 | 0.99 - 1.02 | 0.386 | | | |
| Total physical activity scores | 1.00 | 1.00 - 1.00 | 0.571 | | | |
| **Dietary measures** | | | | | | |
| Lutein, *mg/d* | 0.98 | 0.95 - 1.01 | 0.226 | | | |
| Zeaxanthin, *mg/d* | 0.36 | 0.08 - 1.66 | 0.19 | 0.32 | 0.05 - 0.93 | 0.172 |
| Lutein+Zeaxanthin, *mg/d* | 0.98 | 0.95 - 1.01 | 0.224 | | | |
| *Regular vitamins or supplement* | | | | | | |
| Vitamins or supplement use | 1.8 | 0.88 - 3.69 | 0.108 | 1.86 | 0.82 - 4.05 | 0.127 |
| No vitamins or supplement | Ref | | | | | |
| **Anthropometric measures** | | | | | | |
| Fat Mass or Body Fat (FM) | 1.00 | 0.99 - 1.01 | 0.621 | | | |
| Fat Mass Index in kg/m$^2$ (FMI) | 1.01 | 0.98 - 1.03 | 0.674 | | | |
| Visceral Fat (FT) | 1.06 | 0.96 - 1.18 | 0.255 | | | |
| Lean Mass (LM) | 1.01 | 0.99 - 1.02 | 0.428 | | | |
| Lean Mass Index in kg/m$^2$ (LMI) | 1.03 | 0.97 - 1.09 | 0.359 | | | |
| Muscle Mass (MM) | 1.01 | 0.98 - 1.03 | 0.623 | | | |
| Waist circumference | 1.02 | 1.00 - 1.05 | 0.099 | 1.02 | 0.98 - 1.06 | 0.312 |
| BMI in kg/m$^2$ | 1.01 | 0.99 - 1.03 | 0.592 | | | |
| *BMI in kg/m$^{2\ Cat}$* | | | | | | |
| Healthy | 0.65 | 0.18 - 1.82 | 0.457 | | | |
| Underweight | Ref | | | | | |
| Overweight | 1.09 | 0.55 - 2.09 | 0.8 | | | |
| Obese | 2.7 | 1.06 - 6.57 | **0.032** | 0.8 | 0.18 - 3.26 | 0.763 |
| **Visual performance measures** | | | | | | |
| *Best corrected visual acuity (BCVA)* | | | | | | |
| Oculus dexter | 0.76 | 0.04 - 12.86 | 0.846 | | | |
| Oculus sinister | 0.72 | 0.08 - 6.95 | 0.779 | | | |
| *Contrast sensitivity function* | | | | | | |
| Oculus dexter | 0.73 | 0.00 - 6541.35 | 0.947 | | | |
| Oculus sinister | 1.95 | 0.01 - 336.22 | 0.798 | | | |
| *Macular pigment optical density (MPOD)* | | | | | | |
| MPOD at 1° | 0.35 | 0.06 - 2.12 | 0.251 | | | |
| MPOD at 0.5° | 0.39 | 0.07 - 2.12 | 0.274 | | | |

*(Continued)*

Table 3. (Continued)

| Variable | Bivariate ordinal logistic regression | | | Multivariate ordinal regression | | |
|---|---|---|---|---|---|---|
| | COR | 95% CI (Lower – Upper) | p-value | AOR | 95% CI (Lower – Upper) | p-value |
| *Subjective visual function* | | | | | | |
| VFQ-25 | 0.94 | 0.91 - 0.97 | **<0.001** | 0.95 | 0.91 - 0.99 | **0.008** |
| **Neuropsychological function** | | | | | | |
| MMSE | 0.94 | 0.80 - 1.10 | 0.408 | | | |
| FAS | 0.99 | 0.97 - 1.01 | 0.383 | | | |
| Animal fluency | 1.04 | 0.97 - 1.12 | 0.22 | | | |

Dependent Variable, Psychological Distress on ordinal scale. COR, Odds ratio; CI, Confidence Interval; AOR, Adjusted Odds Ratio; VFQ-25, The National Eye Institute 25-Item Visual Function Questionnaire; MMSE, Mini-Mental State Examination; FAS, F-A-S phonemic verbal fluency test. Bivariate ordinal logistic regression at Wald test ≤ 0.2 considered for inclusion in the multivariate model; Statistical significance set at p ≤ 0.05.

50% of the day outside (AOR = 2.40, 95% CI: 0.97–5.81, p = 0.054) showed a trend toward increased odds, though it was marginally non-significant. Longer swimming duration (AOR = 1.04, 95% CI: 1.01–1.09, p = 0.037) was also associated with a slight increase in odds of higher severity of psychological distress. Conversely, higher VFQ-25 scores (AOR = 0.95, 95% CI: 0.91–0.99, p = 0.008) were associated with a reduction in the odds of experiencing severe psychological distress, suggesting that better visual function may be protective. Other variables, such as ethnicity, protective eyewear, BMI category, and dietary intake of lutein and zeaxanthin, were not statistically significant after adjustment (p > 0.05), suggesting their effects may be confounded by other covariates. Overall, these findings highlight that certain behavioral and visual function factors can increase or reduce the odds of the severity of psychological distress, whereas many demographic, health, and anthropometric measures appear less influential in mediating psychological distress in this population (see **Table 3**).

## Discussion

This novel study examined psychological distress and its lifestyle, dietary, visual, and cognitive correlates among young adults in Ghana. We conjectured that lifestyle behaviors, dietary intake, visual function, and cognitive factors would be significantly associated with the severity of psychological distress. Overall, 23.26% of participants experienced psychological distress, indicating that nearly one in five were affected. Those with psychological distress generally had lower intakes of lutein and zeaxanthin and were less likely to wear eyeglasses, suggesting potential links between visual and mental health. Participants who consumed alcohol occasionally or spent more time swimming were more likely to experience higher severity of psychological distress, whereas those with better subjective visual function, as indicated by higher NEI-VFQ-25 scores, were less likely to experience psychological distress. Spending more than half of the day outdoors showed a marginal tendency toward increased risk of psychological distress in this sample (see **Fig 3**).

Optimal mental health is vital for overall well-being and effective daily functioning [48]. Young adults in Ghana constitute a major demographic group and are at a developmental stage characterized by shifting living arrangements, evolving independence, and changing lifestyle habits, including diet and substance use. Reduced parental oversight and increased exposure to stressors such as academic pressure and financial uncertainty may elevate susceptibility to mental health challenges, including psychological distress [8]. Concurrently, this phase typically aligns with higher education, which brings intense academic demands that may strain the visual system and impact eye health [8]. Although national data on psychological distress in this population are limited, anecdotal reports of increasing suicide rates among Ghanaian university students underscore the need for greater attention to mental health within this age group. To this end, we hypothesize that psychological distress is common among healthy young adults and is associated with modifiable lifestyle and behavioral factors.

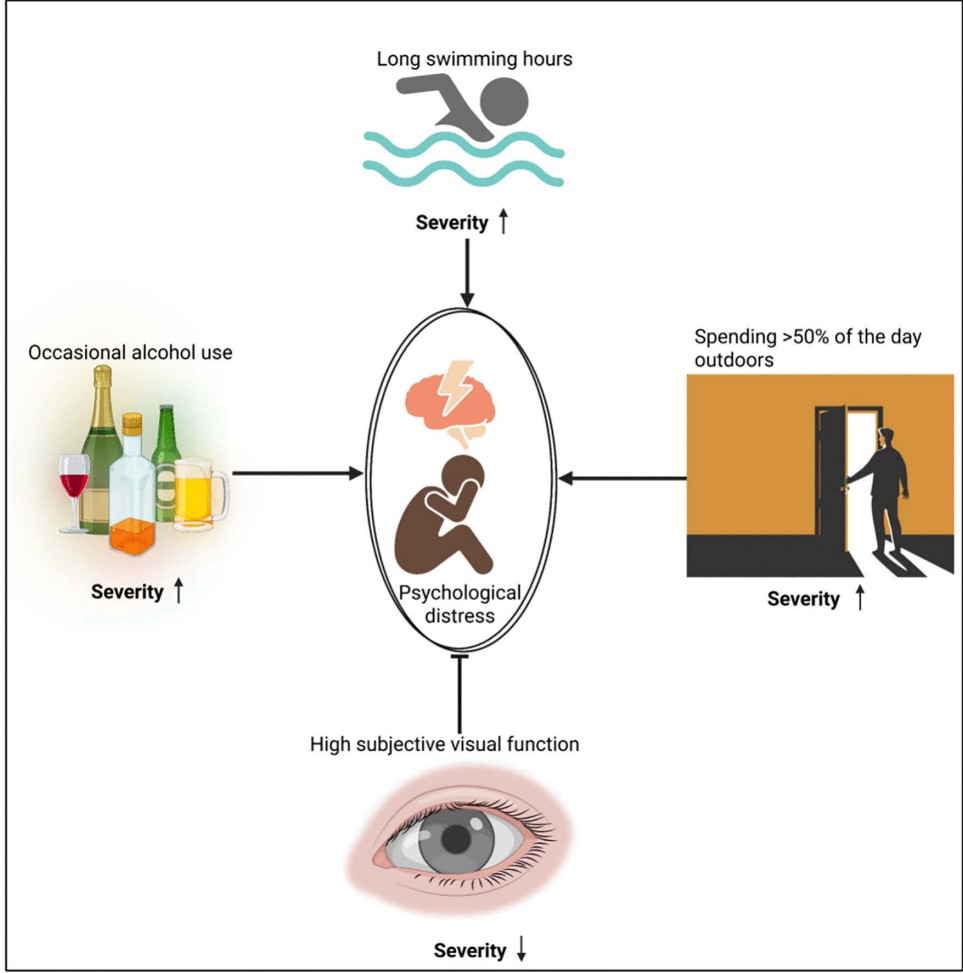

**Fig 3. Concept map of main contributors to PSD.** The figure illustrates factors influencing psychological distress severity among young adults. On the **left**, occasional alcohol use is associated with increased severity; at the **top**, long swimming hours also heighten severity; on the **right**, spending more than half the day outdoors increases severity; and at the **bottom**, high subjective visual function is linked to reduced severity. Created in BioRender. Osei Duah, I. (2025) https://BioRender.com/ c9a1al6c9a1al6.

In this study, approximately one in five participants reported symptoms indicative of mild to severe psychological distress. This prevalence is consistent with estimates from previous studies conducted in Africa [49,50], Australia [51,52], Asia [53–55], Europe [56,57] and America [58] where reported rates range between 15% and 60% [49–58]. These findings reinforce evidence that young adults constitute a population subgroup at heightened risk for psychological distress. The distribution of symptom severity underscores the need for early identification, preventive screening, and timely psychological interventions to mitigate symptom progression and enhance coping capacity among vulnerable individuals.

The observation that participants experiencing psychological distress reported lower dietary intakes of lutein and zeaxanthin, accompanied by a trend toward decreased MPOD, suggests a potential link between psychological health and retinal carotenoid status (see Fig 1). These xanthophyll carotenoids, primarily derived from fruits and vegetables, accumulate in the retina and brain and are known to support visual and cognitive functioning [14,15,20]. Previous studies have linked reduced levels of these nutrients to increased psychological stress [12]. Lutein and zeaxanthin are carotenoids

with antioxidant and anti-inflammatory properties that support both retinal and neural function [59]. Deficiencies in these nutrients may impair visual processing and increase oxidative stress in the brain, potentially contributing to mood disturbances and emotional dysregulation [12]. However, when psychological distress is assessed according to severity rather than a simple presence/absence, the differences in lutein and zeaxanthin intake are attenuated. This attenuation may reflect the multifactorial nature of more severe distress, where dietary factors are only one of many contributors, alongside genetic predisposition, environmental stressors, and lifestyle behaviors (see Table 2). Collectively, these findings highlight the potential value of nutrition-based interventions and educational programs aimed at promoting both visual and mental health among young adults [12].

Unexpectedly, a greater proportion of participants without psychological distress reported wearing eyeglasses. This finding suggests that vision correction may exert a subtle but protective effect on psychological well-being. Supporting this, Guan *et al.* [60] found that myopic students who studied intensively experienced reductions in learning- and physically-related anxiety after being provided with spectacles, underscoring the role of optimal visual function in cognitive comfort and emotional regulation. Biologically, uncorrected refractive errors can induce visual disturbances that alter the perception of the light–dark cycle, leading to circadian rhythm disruption [61]. Such disruptions are closely linked to mood dysregulation and mental distress [62]. Given the substantial visual and cognitive demands faced by young adults, routine vision screening and timely correction of refractive errors could serve as practical interventions to mitigate psychological distress in high-risk groups [60].

Alcohol consumption was also found to be positively associated with increased severity of psychological distress, consistent with findings by Balogun *et al.* [63] and Geisner *et al.* [64], who observed greater distress among individuals with heavier or more frequent alcohol use [63,64]. From a psychobiological perspective, alcohol may initially act as a maladaptive coping mechanism, providing short-term mood relief while disrupting neurotransmitter balance (such as serotonin, dopamine, and GABA) and exacerbating stress reactivity and sleep disturbances over time. These neurochemical alterations can increase vulnerability to mood dysregulation and anxiety symptoms. Consequently, interventions aimed at reducing alcohol intake or increasing abstinent days may have measurable benefits for mental health in this population [64].

Participants who spent more time swimming were more likely to experience higher severity of psychological distress. While swimming is widely regarded as a therapeutic activity with calming effects that promote mental well-being and inner peace, this finding suggests that increased swimming time does not always correspond with better mental health outcomes [65]. It is possible that individuals experiencing higher levels of psychological distress may use swimming as a coping mechanism, resulting in more frequent or prolonged engagement. Alternatively, excessive time spent swimming could reflect lifestyle imbalances, social isolation, or other stress-related behaviors that exacerbate psychological distress. These results underscore the complex relationship between recreational activities and mental health, highlighting that the amount of engagement alone does not necessarily indicate improved psychological well-being.

Further, participants with better self-reported visual function, as indicated by higher NEI-VFQ-25 scores, were less likely to experience psychological distress. This finding aligns with previous studies showing that self-perceived vision impairment is associated with higher emotional distress and depressive symptoms [24,66]. Interestingly, this association was observed for subjective, rather than objective, visual measures. One possible explanation is that self-reported vision reflects not only measurable visual acuity but also the functional and psychosocial impact of vision on daily life, including the ability to engage in work, study, or social activities [24,66,67] (see Fig 2). In contrast, objective measures may not fully capture the practical limitations or frustrations individuals experience, which could influence psychological well-being [24]. Since participants without mental distress were more likely to wear spectacles, early detection and correction of vision problems could represent an overlooked, yet biologically plausible, avenue for preventing or alleviating psychological distress among young adults (see Table 2). Accordingly, we recommend routine eye care screening to detect early vision difficulties that may contribute to psychological distress in this population.

Finally, participants who spent more than half of their day outdoors exhibited a marginal tendency toward higher psychological distress in this sample. While exposure to natural environments is often associated with mental health benefits

[68,69], these findings suggest that extended outdoor time did not necessarily confer protection against psychological distress for this group. It is possible that those experiencing greater distress may have spent more time outdoors as a coping strategy, or that other contextual factors, such as social interactions, environmental stressors, or the nature of outdoor activities, influenced this association [70,71]. Although the effect was modest, it highlights the complex relationship between time spent outdoors and mental health outcomes [71].

### Strengths and limitation

Factors contributing to psychological distress among young adults in Ghana remain unclear. This investigation provides evidence of the prevalence of psychological distress among this population. The study generates prevalence data that can contribute to systematic reviews and inform future longitudinal research. High-risk behaviors and modifiable factors associated with psychological distress were also identified, which may inform strategies to reduce its burden. These findings have potential implications for institutional and national intervention policies and may guide the development of coping strategies for populations at higher risk of psychological distress. Regarding sample representativeness, participants were recruited from a university population using systematic random sampling. The University offer access to a cross-section of young adults from diverse backgrounds in terms of age, sex, socioeconomic status, and educational attainment. While the sample captures key demographic characteristics of the target population, some subgroups may be underrepresented, which could limit generalizability. Nevertheless, the systematic recruitment approach supports external validity, and the findings are likely applicable to broader young adult populations in Ghana and similar contexts. However, the study has some limitations. Lutein and zeaxanthin intake was assessed using dietary recall, which may not fully represent actual intake due to potential recall bias. This method, however, has been demonstrated to provide estimates consistent with plasma carotenoid measurements and is commonly used in clinical and epidemiological research [72]. Intimate partner relationships are common among young adults and may influence mental health outcomes. Although our study did not investigate this, future research assessing their impact could clarify some of the observed associations and further inform strategies to mitigate psychological distress [73]. Additionally, due to the relatively small number of participants reporting regular alcohol use and the variability across consumption categories, we were unable to conduct formal dose–response analyses. Future studies should aim to recruit larger samples of participants who consume alcohol frequently to allow for more robust dose–response investigations and clearer understanding of potential associations with psychological distress.

### Conclusion

Among young adults in this study, approximately one in five experienced psychological distress, highlighting a substantial mental health burden. Those affected generally reported low intakes of xanthophyll carotenoids (lutein and zeaxanthin) and lower use of corrective eyewear, while alcohol consumption was linked to higher likelihood of distress. Conversely, better self-reported vision and higher NEI-VFQ-25 scores appeared protective. Participants who consumed alcohol occasionally or spent more time swimming were more likely to experience greater severity of psychological distress, whereas better visual function mitigated this risk. Spending more than half of the day outdoors showed a marginal tendency toward increased distress, suggesting complex interactions between lifestyle behaviors and mental health. Collectively, these findings underscore the potential benefits of integrated interventions that combine mental health screening with eye care services, dietary education, and substance use counseling to optimize carotenoid intake and visual health while reducing alcohol consumption, thereby lowering the risk of psychological distress among young adults in Ghana.

### Supporting information

**S1 Table. Dataset on integrative examination of PSD.**
(XLSX)

## Acknowledgments

The authors thank Professor Billy Wooten, Department of Psychology, Brown University, Providence, RI, USA, for the generous donation of a macular densitometer. We are indebted to Dr. Micheal Gyawu-Odame, Dr. Stephannie Obiri-Yeboah, and Dr. Stacy Ewurama Horthman of the Department of Optometry and Visual Science, College of Science, Kwame Nkrumah University of Science and Technology (KNUST), Kumasi, Ghana, for their assistance with data collection. We also acknowledge Dr. Reginald Acquah Gyamfi and Samuel Andy Afari of the same department for technical assistance. The authors further acknowledge Bernice Lebene Tettey of the Department of Chemistry, KNUST, Kumasi, Ghana, for providing technical assistance on Fig 3. Finally, we are grateful to the research subjects for their willingness to participate.

## Author contributions

**Conceptualization:** Isaiah Osei Duah Junior.

**Data curation:** Isaiah Osei Duah Junior, Wendy Ofori Asare, Elizabeth J. Johnson.

**Formal analysis:** Isaiah Osei Duah Junior, Wendy Ofori Asare, Elizabeth J. Johnson, Kwadwo Owusu Akuffo.

**Investigation:** Isaiah Osei Duah Junior, Wendy Ofori Asare, Elizabeth J. Johnson, Charllote Boateng, Hubert Osei Acheampong, David Ben Kumah, Kwadwo Owusu Akuffo.

**Methodology:** Isaiah Osei Duah Junior, Wendy Ofori Asare, Elizabeth J. Johnson, Charllote Boateng, Hubert Osei Acheampong, David Ben Kumah, Kwadwo Owusu Akuffo.

**Project administration:** Isaiah Osei Duah Junior, Wendy Ofori Asare, Elizabeth J. Johnson, Charllote Boateng, Hubert Osei Acheampong, David Ben Kumah, Kwadwo Owusu Akuffo.

**Resources:** Isaiah Osei Duah Junior, Wendy Ofori Asare, Elizabeth J. Johnson, Hubert Osei Acheampong, David Ben Kumah, Kwadwo Owusu Akuffo.

**Software:** Isaiah Osei Duah Junior, Wendy Ofori Asare, Elizabeth J. Johnson, David Ben Kumah, Kwadwo Owusu Akuffo.

**Supervision:** Isaiah Osei Duah Junior, Kwadwo Owusu Akuffo.

**Validation:** Isaiah Osei Duah Junior, Elizabeth J. Johnson, Kwadwo Owusu Akuffo.

**Visualization:** Isaiah Osei Duah Junior, Elizabeth J. Johnson, Kwadwo Owusu Akuffo.

**Writing – original draft:** Isaiah Osei Duah Junior, Wendy Ofori Asare, Elizabeth J. Johnson, Kwadwo Owusu Akuffo.

**Writing – review & editing:** Isaiah Osei Duah Junior, Elizabeth J. Johnson, Charllote Boateng, Hubert Osei Acheampong, David Ben Kumah, Kwadwo Owusu Akuffo.

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
