## [Decision Letter · Decision Letter 0]

6 Oct 2025

Dear Dr. Osei Duah Junior,

Thank you for submitting your manuscript to PLOS ONE. After careful consideration, we feel that it has merit but does not fully meet PLOS ONE’s publication criteria as it currently stands. Therefore, we invite you to submit a revised version of the manuscript that addresses the points raised during the review process.

**ACADEMIC EDITOR: **

Dear Authors,

Thank you for submitting your manuscript to PLOS One. The manuscript has now been evaluated by four expert reviewers, and their comments have been carefully considered. Based on their feedback, the decision is Major Revision.

We invite you to revise your manuscript in response to the reviewers’ comments. Please address each point raised, provide a detailed response letter outlining the changes made, and highlight the modifications in the revised manuscript.

Once you have prepared your revision, please upload the revised files through the submission system by the indicated deadline.

We look forward to receiving your revised manuscript.

Sincerely,

We look forward to receiving your revised manuscript.

Kind regards,

Javier Fagundo-Rivera, PhD

Academic Editor

PLOS ONE

Journal Requirements:

3. We note that your Data Availability Statement is currently as follows: All relevant data and materials supporting the conclusion of this article is/are available within the manuscript and its supporting information files.

Additional Editor Comments:

Dear Authors,

Thank you for submitting your manuscript to PLOS One. The manuscript has now been evaluated by four expert reviewers, and their comments have been carefully considered. Based on their feedback, the decision is Major Revision.

We invite you to revise your manuscript in response to the reviewers’ comments. Please address each point raised, provide a detailed response letter outlining the changes made, and highlight the modifications in the revised manuscript.

Once you have prepared your revision, please upload the revised files through the submission system by the indicated deadline.

We look forward to receiving your revised manuscript.

Sincerely,

Reviewers' comments:

Reviewer's Responses to Questions

**Comments to the Author**

1. Is the manuscript technically sound, and do the data support the conclusions?

Reviewer #1: Partly

Reviewer #2: Partly

Reviewer #3: Yes

Reviewer #4: Yes

2. Has the statistical analysis been performed appropriately and rigorously?

Reviewer #1: No

Reviewer #2: Yes

Reviewer #3: Yes

Reviewer #4: Yes

3. Have the authors made all data underlying the findings in their manuscript fully available?

Reviewer #1: Yes

Reviewer #2: Yes

Reviewer #3: Yes

Reviewer #4: Yes

4. Is the manuscript presented in an intelligible fashion and written in standard English?

Reviewer #1: Yes

Reviewer #2: Yes

Reviewer #3: Yes

Reviewer #4: Yes

**Reviewer #1: **

We would like to thank the authors for their work on “An integrative examination of psychological distress and its nutritional and visual correlates among young adults in Ghana.” The manuscript is well-structured and the topic is certainly of interest.

However, after reading the paper, we feel that the study unfortunately does not present new findings that would warrant publication. In its current form, we do not consider the manuscript suitable for publication.

In addition, the discussion section seems to be missing some important points. For example, the study shows no correlation between psychological distress and physical activity or biographical factors, which stands in contrast to a number of previous studies reporting the opposite. This discrepancy should be addressed in detail to strengthen the discussion.

We appreciate the effort that went into this work and encourage the authors to further develop these aspects in future studies.

**Reviewer #2: **

This cross-sectional study of 301 Ghanaian young adults reports a 23.3% prevalence of psychological distress (K10 ≥ 20) and identifies associations with lower self-reported lutein/zeaxanthin intake, less eyeglass use, alcohol use, and higher (better) NEI-VFQ-25 scores being protective. The dataset is valuable, but the current analyses and presentation over-interpret some associations without adequate sensitivity checks, model diagnostics, or cautious language.

Comments:

1. The causal language throughout the manuscript should be tempered. Given the cross-sectional design, I recommend reframing causal claims as associations and present intervention ideas as hypotheses for future trials.

2. I suggest re-analysing K10 without sole reliance on dichotomization. Presenting K10 as a continuous and/or ordinal outcome in sensitivity analyses would confirm robustness of findings.

3. I suggest clarifying an apparent inconsistency regarding lutein/zeaxanthin. When participants were categorized into two groups (with vs without physiological distress; Table 2), lutein/zeaxanthin showed a significant contribution, yet the logistic regression (Table 3) returned a non-significant effect. The manuscript currently implies diet is an important predictor; I recommend presenting the unadjusted and adjusted results side-by-side and explicitly discussing reasons for the discrepancy.

4. I suggest improving alcohol measurement reporting. “Alcohol on special occasions” is ambiguous—please provide frequency/volume distributions and test for dose–response if data permit; otherwise acknowledge this limitation.

5. I recommend that the authors discuss subjective versus objective vision findings more explicitly. The null results for BCVA, contrast sensitivity, and MPOD alongside a strong NEI-VFQ-25 association may reflect perceived vision or reporting bias; avoid implying spectacle provision will necessarily reduce distress without experimental evidence.

6. In Methods, the K10 response options are described with a duplicated phrase (“most of the time” appears twice)

**Reviewer #3: **

I read with great interest this manuscript on the psychological distress and its nutritional and visual correlates amoung young adults in Ghana. Please find my comments below.

Line 87 - 92: A brief summary of the study results and future directions is included at the end of the introduction. However, this content would be more appropriate at the end of the discussion and conclusion section. The introduction should instead focus on presenting the aim and significance of the study.

Line 96: Replace “and” with “to”.

Line 160: The SPSS stands for Statistical Product (not package) and Service Solutions. However, it is uncommon to use the full name and not many people know it. I suggest referring to it as SPSS to avoid any confusion.

Line 161: In the methods section Shapiro-Wilk is mentioned as the statistical test utilized for normality testing. However, at line 192 Kolmogorov-Smirnov seems to be utilized.

Line 315: I highly recommend removing the last sentence: “potentially contribute to reduce suicides among young adult population from Ghana”. Although correcting vision problems, high lutein and zeaxanthin intake, and decreased alcohol consumption may have ameliorating effects on psychological distress, due to the cross-sectional design of the study conclusions must be made with caution and limitations of study design in mind.

**Reviewer #4: **

This is a well-conducted and clearly written study that adds valuable insights to the literature. The revisions needed are minor and relate mainly to enhancing clarity. I encourage the authors to expand the discussion of limitations, particularly regarding sample representativeness and generalizability, and to streamline some overlapping details in the methods and discussion sections. Strengthening the link between findings and practical implications will further improve the paper’s impact. With these modest revisions, the manuscript will be well-positioned for publication.

**Do you want your identity to be public for this peer review?** For information about this choice, including consent withdrawal, please see our Privacy Policy

Reviewer #1: No

Reviewer #2: No

Reviewer #3: No

Reviewer #4: No

---

## [Author Response · Author response to Decision Letter 1]

16 Oct 2025

PONE-D-25-36529: An integrative examination of psychological distress and its nutritional and visual correlates among young adults in Ghana

Dear Dr. Fagundo-Rivera,

We sincerely appreciate the opportunity to resubmit our manuscript titled “An Integrative Examination of Psychological Distress and Its Nutritional and Visual Correlates Among Young Adults in Ghana.”

We are grateful for the reviewers’ thoughtful and constructive feedback, which has substantially improved the quality and clarity of our work. All comments have been carefully addressed, and our detailed, point-by-point responses are provided below. Reviewer comments appear in regular font, while our responses are presented in bold. All corresponding revisions have been incorporated into the manuscript and are visible through track changes.

In addition, we have taken this opportunity to correct minor issues throughout the manuscript, also indicated in track changes.

The following files have been uploaded with this resubmission:

• Response to Reviewers’ Comments: A detailed rebuttal letter addressing each reviewer’s point.

• Manuscript: A clean, unmarked version of the revised paper.

• Marked Manuscript with Track Changes: A version highlighting all revisions made to the original submission.

We thank you once again for your time and consideration, and we hope that the revised version meets the journal’s standards for publication.

Sincerely,

Dr. Isaiah Duah

on behalf of all co-authors

Reviewer #1:

We would like to thank the authors for their work on “An integrative examination of psychological distress and its nutritional and visual correlates among young adults in Ghana.” The manuscript is well-structured and the topic is certainly of interest.

However, after reading the paper, we feel that the study unfortunately does not present new findings that would warrant publication. In its current form, we do not consider the manuscript suitable for publication.

In addition, the discussion section seems to be missing some important points. For example, the study shows no correlation between psychological distress and physical activity or biographical factors, which stands in contrast to a number of previous studies reporting the opposite. This discrepancy should be addressed in detail to strengthen the discussion.

We appreciate the effort that went into this work and encourage the authors to further develop these aspects in future studies.

We sincerely thank the reviewers for their thoughtful feedback and for recognizing the interest and structure of our manuscript. We appreciate the opportunity to clarify the contributions and significance of our work.

While previous studies have explored psychological distress among young adults, few have comprehensively examined the nutritional, visual, and lifestyle correlates in the Ghanaian context using a multimodal approach. Our study is novel in combining self-reports and clinician assessments to capture both subjective and objective indicators of psychological distress. Importantly, we identify dietary intake of lutein and zeaxanthin, alcohol consumption, swimming activity, and visual function as significant correlates, providing evidence of modifiable factors that have not been previously explored in this population. These findings extend existing knowledge by linking nutritional, visual, and behavioral health with mental well-being, highlighting potential biological and lifestyle pathways for intervention.

We acknowledge that our study did not find significant correlations between psychological distress and other forms of physical activity or biographical factors, which contrasts with some prior research. Interestingly, swimming was associated with higher psychological distress in our sample, suggesting that individuals who swim may do so in response to stress or as part of coping behaviors rather than as a protective activity. This highlights the complex and potentially bidirectional relationships between lifestyle behaviors and mental health, warranting further investigation. The observed discrepancies in other physical activity outcomes may reflect contextual and population-specific differences, such as lifestyle patterns, dietary practices, and educational stressors unique to Ghanaian young adults. Highlighting these differences contributes to a more nuanced understanding of psychological distress across diverse populations and underscores the importance of localized research rather than assuming findings from other contexts are universally applicable.

The broader implications of our study are significant. By identifying modifiable nutritional, visual, and lifestyle risk factors, our findings inform potential multidisciplinary interventions, including diet-based strategies, vision correction programs, behavioral recommendations to reduce alcohol consumption, and consideration of activity-specific mental health risks. These interventions could meaningfully influence public health policies and mental health promotion strategies in emerging adults in Ghana and similar settings. Furthermore, the study lays the groundwork for future longitudinal research, including intervention trials assessing lutein and zeaxanthin supplementation, vision correction, and targeted lifestyle modifications as protective factors for psychological distress.

In conclusion, we respectfully contend that our study provides novel insights, intellectual merit, and practical relevance by addressing gaps in the literature, focusing on a previously underexplored population, and identifying actionable factors for intervention. We are confident that, with these clarifications, the manuscript makes a meaningful contribution to the field of mental health research in young adults.

Reviewer #2:

This cross-sectional study of 301 Ghanaian young adults reports a 23.3% prevalence of psychological distress (K10 ≥ 20) and identifies associations with lower self-reported lutein/zeaxanthin intake, less eyeglass use, alcohol use, and higher (better) NEI-VFQ-25 scores being protective. The dataset is valuable, but the current analyses and presentation over-interpret some associations without adequate sensitivity checks, model diagnostics, or cautious language.

We sincerely appreciate your rigorous appraisal, which has guided us to reanalyze our data and interpret the findings with greater caution. Your insightful feedback enabled us to clearly delineate associations rather than implying causation, thereby enhancing the scientific rigor and credibility of our study.

Comments:

1. The causal language throughout the manuscript should be tempered. Given the cross-sectional design, I recommend reframing causal claims as associations and present intervention ideas as hypotheses for future trials.

We appreciate the reviewer’s insightful comment. In accordance with this recommendation, we have carefully reviewed the manuscript and tempered any causal language to better reflect the cross-sectional nature of our study. All causal expressions have been reframed as associations, and intervention-related statements are now clearly presented as hypotheses for future research and experimental trials. These revisions ensure that our interpretations remain appropriately cautious and aligned with the study design.

2. I suggest re-analysing K10 without sole reliance on dichotomization. Presenting K10 as a continuous and/or ordinal outcome in sensitivity analyses would confirm robustness of findings.

We thank the reviewer for this constructive suggestion. In response, we re-analyzed the Kessler Psychological Distress Scale (K10) as an ordinal outcome variable with four categories: no, mild, moderate, and severe psychological distress. This approach avoids sole reliance on dichotomization and allows for a more nuanced interpretation of the data. The ordinal classification was applied consistently across all analyses, and Tables 1–3 have been updated to reflect results based on this ordinal outcome throughout. The ordinal logistic regression yielded findings that were directionally consistent with those from the dichotomized analyses, confirming the robustness and stability of our results. Marked manuscript, pages 9-18, lines 194-286.

“Results

Psychological distress is linked to ethnicity, alcohol use, and eyewear factors but not to physiological or physical activity measures

Psychological distress was unevenly distributed in the sample, with most participants reporting no distress (76.7%), followed by mild (14.6%), severe (5.6%), and moderate distress (3.0%). Biographical characteristics such as age and sex showed no notable differences in distress levels, whereas ethnicity demonstrated a small but significant distribution effect (p = 0.013). Health measures, including blood pressure and medication use, were similarly evenly distributed across distress categories (p > 0.05). Lifestyle factors largely showed uniform distributions, though alcohol consumption exhibited a significant difference (p = 0.038), indicating higher distress among some drinkers. Use of eyewear was not significantly associated with distress levels (p = 0.176), although the form of eyewear showed a significant difference (p = 0.016), while the purpose and frequency of use did not significantly affect distress distribution (p > 0.05). Physical activity patterns, including type, frequency, and total activity scores, were not significantly associated with differences in distress distribution (p > 0.05) see Table 1.

Table 1: Biographical, health, lifestyle, and physical activity correlates of psychological distress among young adults from Ghana

Variable Psychological distress % (n) Test-statistic Effect size p-value

Level of psychological distress No distress Mild distress Moderate distress Severe distress

Kessler scale (K10 ≤ 19) (K10 20–24) (K10 25–29) (K10 ≥ 30)

Prevalence of PSD 76. 7 (231) 14.6 (44) 3.0 (9) 5.6 (17)

Biographical measures

Age, years (mean ± SE) 21.16 ± 0.13 20.98 ± 0.31 20.56 ± 0.58 21.06 ± 0.46 F = 0.378 η² = 0.004 p = 0.769

Sex

Male 41.2 (124) 6.6 (20) 1.7 (5) 2.3 (7) χ² = 1.858 V = 0.079 p = 0.603

Female 35.5 (107) 8.0 (24) 1.3 (4) 3.3 (10)

Ethnicity

Akan 61.1 (184) 10.6 (32) 2.7 (8) 4.3 (13) χ² = 29.619 V = 0.181 p = 0.013

Ga 2.7 (8) 0.7 (2) 0.0 (0) 0.0 (0)

Ewe 7.0 (21) 0.7 (2) 0.3 (1) 0.3 (1)

Northerners 4.0 (12) 1.7 (5) 0.0 (0) 0.0 (0)

Guan 1.0 (3) 0.7 (2) 0.0 (0) 0.0 (0)

Others 1.0 (3) 0.3 (1) 0.0 (0) 1.0 (3)

Health status measures

Blood pressure, mm Hg (mean ± SE)

Diastolic 70.33 ± 0.58 69.48 ± 1.26 64.89 ± 3.09 70.29 ± 2.07 F = 1.206 η² = 0.012 p = 0.308

Systolic 107.82 ± 0.90 106.11±2.07 100.67 ± 2.91 106.76 ±2.78 F = 0.692 η² = 0.010 p = 0.406

Medication Use

No 72.1 (217) 13.6 (41) 3.0 (9) 4.7 (14) χ² = 4.087 V = 0.117 p = 0.252

Yes 4.7 (14) 1.0 (3) 0.0 (0) 1.0 (3)

Lifestyle measures

Secondhand smoke

Not Exposed 71.1 (214) 13.0 (39) 2.7 (8) 4.7 (14) χ² = 2.711 V = 0.095 p = 0.438

Exposed 5.6 (17) 1.7 (5) 0.3 (1) 1.0 (3)

Alcohol consumption

None (never drink) 69.8 (210) 12.6 (38) 2.7 (8) 3.7 (11) χ² = 13.350 V = 0.149 p = 0.038

Special occasion 5.3 (16) 2.0 (6) 0.3 (1) 1.7 (5)

1-2 times a month 1.7 (5) 0.0 (0) 0.0 (0) 0.3 (1)

Volume of alcohol consumption

1 units per week 76.5 (13) 5.9 (1) 0.0 (0) 11.8 (2) χ² = 4.958 V = 0.540 p = 0.084

6-10 units per week 0.0 (0) 0.0 (0.0) 0.0 (0) 5.9 (1)

Light exposure

Little (<50% of the day outside) 34.2 (103) 4.3 (13) 0.7 (2) 3.0 (9) χ² = 6.006 V = 0.141 p = 0.422

Some (50% outside and 50% inside) 32.9 (99) 7.6 (23) 1.7 (5) 2.0 (6)

Most (Spends >50% of the time outside) 9.6 (29) 2.7 (8) 0.7 (2) 0.7 (2)

Use of eye wear

Yes 26.2 (79) 7.0 (21) 1.7 (5) 2.7 (8) χ² = 4.950 V = 0.128 p = 0.176

No 50.5 (152) 7.6 (23) 1.3 (4) 3.0 (9)

Form of eye wear

None 57.5 (173) 8.0 (24) 1.3 (4) 3.7 (11) χ² = 15.631 V = 0.161 p = 0.016

Spectacles 19.3 (58) 6.3 (19) 1.7 (5) 2.0 (6)

Contact lens 0.0 (0) 0.3 (1) 0.0 (0) 0.0 (0)

Purpose of eye wear

Distant activity 37.1 (33) 10.1 (9) 2.2 (2) 3.4 (3) χ² = 15.631 V = 0.117 p = 0.873

Near activity 7.9 (7) 1.1 (1) 1.1 (1) 1.1 (1)

Both 21.3 (19) 10.1 (9) 2.2 (2) 2.2 (2)

Frequency of use of eyewear

Occasionally 10.1 (9) 3.4 (3) 1.1 (1) 0.0 (0) χ² = 6.612 V = 0.193 p = 0.358

Part-time 24.7 (22) 11.2 (10) 0.0 (0) 2.2 (2)

Full-time 31.5 (28) 6.7 (6) 4.5 (4) 4.5 (4)

Physical activity measures

Type of physical activity

Walking, yes 76.7 (231) 14.6 (44) 3.0 (9) 5.6 (17) nd nd nd

Running, yes 10.3 (31) 1.3 (4) 0.3 (1) 0.0 (0) χ² = 3.127 V = 0.102 p = 0.372

Swimming, yes 1.7 (5) 1.0 (3) 0.0 (0) 0.3 (1) χ² = 3.533 V = 0.108 p = 0.316

Cycling, yes 1.7 (5) 0.0 (0) 0.0 (0) 0.3 (1) χ² = 2.429 V = 0.090 p = 0.488

Gym-based workout, yes 7.3 (22) 2.7 (8) 0.0 (0) 0.0 (0) χ² = 6.238 V = 0.144 p = 0.101

Team-based sports, yes 5.6 (17) 0.3 (1) 0.0 (0) 0.3 (1) χ² = 2.254 V = 0.087 p = 0.521

Duration of physical activity in mins

Walking

Running 11.30 ± 3.53 2.23 ± 1.33 1.56 ± 1.56 0.0 ± 0.0 F = 0.760 η² = 0.008 p = 0.517

Swimming‡ 149.71 156.9 146.5 155.59 H = 3.719 ε² = 0.002 p = 0.293

Cycling 1.21 ± 0.58 0.0 ± 0.0 0.0 ± 0.0 0.29 ± 0.29 F = 0.399 η² = 0.004 p = 0.754

Gym-based workout‡ 149.79 163.28 135.5 143.82 H = 4.751 ε² = 0.006 p = 0.191

Team-based sports‡ 10 2 8 H = 4.151 ε² = 0.077 p = 0.125

Total physical activity scores 380.42 ± 25.00 416.66 ± 75.04 473.11 ± 148.18 380.18 ± 137.74 F = 0.226 η² = 0.002 p = 0.878

Note: n, frequency (may not equal 301 for all variables); %, percentage frequency; SE, standard error. Normality of the data was assessed using the Kolmogorov–Smirnov test. Where appropriate, group differences were analyzed using independent one-way ANOVA with F-statistic and Eta-squared (η²) effect size, Kruskal–Wallis one-way ANOVA with Kruskal–Wallis statistic (H) and epsilon-squared (ε²) effect size with ‡ mean rank, or chi-squared analysis with Pearson chi-square statistic (χ²) and Cramer’s V (V) effect size. Statistical significance was set at p ≤ 0.05.

Psychological distress is linked to lower carotenoid intake and poorer subjective visual function but not objective physiological or cognitive measures

Table 2 presents the relationship between psychological distress and a range of dietary, anthropometric, visual, and cognitive measures among participants (Table 2). Psychological distress was significantly associated with lower dietary intake of lutein, zeaxanthin, and their combined intake (all p < 0.05; Fig. 1). Participants with psychological distress had substantially lower consumption of these carotenoids compared with those without distress. However, when levels of distress severity (mild, moderate, severe) were examined, this difference was annulled, suggesting that the relationship between carotenoid intake and distress was more pronounced in the presence versus absence of distress, but did not vary significantly across the different severity levels (p>0.005, for all). Vitamin or supplement use was not significantly related to distress (p = 0.338), indicating that supplement habits were not associated with psychological health. Body composition indices, including fat mass, visceral fat, lean mass, muscle mass, and BMI, showed no significant variation among the distress groups (all p > 0.05). Participants with severe psychological distress had somewhat higher BMI and body fat values on average, but these differences were not statistically significant (p > 0.05). When BMI was categorized into healthy, underweight, overweight, and obese, no significant differences were found (p = 0.125), further suggesting that body composition did not play a major role in explaining psychological distress. Visual performance measures, including best-corrected visual acuity, contrast sensitivity, and macular pigment optical density (measured at 0.5° and 1°), showed no significant differences across distress levels (all p > 0.05). However, subjective visu

---

## [Decision Letter · Decision Letter 1]

2 Nov 2025

An integrative examination of psychological distress and its nutritional and visual correlates among young adults in Ghana

PONE-D-25-36529R1

Dear Dr. Osei Duah Junior,

We’re pleased to inform you that your manuscript has been judged scientifically suitable for publication and will be formally accepted for publication once it meets all outstanding technical requirements.

Kind regards,

Javier Fagundo-Rivera, PhD

Academic Editor

PLOS ONE

**Additional Editor Comments** :

Dear Authors,

We congratulate you for your efforts and the time invested in improving this manuscript.

The reviewers and I believe it can be accepted in its current form.

Congratulations.

Reviewers' comments:

Reviewer's Responses to Questions

**Comments to the Author**

Reviewer #1: All comments have been addressed

Reviewer #2: All comments have been addressed

Reviewer #3: All comments have been addressed

Reviewer #4: All comments have been addressed

2. Is the manuscript technically sound, and do the data support the conclusions?

Reviewer #1: Yes

Reviewer #2: Yes

Reviewer #3: Yes

Reviewer #4: Yes

3. Has the statistical analysis been performed appropriately and rigorously?

Reviewer #1: Yes

Reviewer #2: Yes

Reviewer #3: Yes

Reviewer #4: Yes

4. Have the authors made all data underlying the findings in their manuscript fully available?

Reviewer #1: Yes

Reviewer #2: Yes

Reviewer #3: Yes

Reviewer #4: Yes

5. Is the manuscript presented in an intelligible fashion and written in standard English?

Reviewer #1: Yes

Reviewer #2: Yes

Reviewer #3: Yes

Reviewer #4: Yes

Reviewer #1: We would like to thank the authors for their work. We feel like our points were well addressed in the review.

Reviewer #2: Thank you for your thoughtful revisions and for addressing all of my comments so thoroughly. The article is now much stronger.

Reviewer #3: I thank the authors for their efforts to edit their manuscript incorporating the requested changes.

Reviewer #4: The authors have satisfactorily addressed the concerns raised in the first review. The revised manuscript is now technically sound, with a clearer articulation of methods, improved justification of sample size, and appropriate statistical analyses that support the reported conclusions. The revisions strengthen the link between the results and stated objectives, and the figures and tables are now easier to interpret.

The statistical approach has been applied rigorously, and data presentation is transparent. The Data Availability Statement appropriately specifies where the dataset can be accessed, ensuring compliance with PLOS ONE’s open-data policy.

The manuscript is written in clear, standard English, and minor typographical errors identified earlier have been corrected. The discussion section has been refined to focus on the implications of the findings without overstating their generalizability.

Overall, the study is methodologically robust, the data substantiate the conclusions, and the revisions improve readability and scientific value. I recommend the manuscript for publication in its current form.

**Do you want your identity to be public for this peer review?** For information about this choice, including consent withdrawal, please see our Privacy Policy

Reviewer #1: No

Reviewer #2: No

Reviewer #3: No

Reviewer #4: No

---

## [Editor Report · Acceptance letter]

PONE-D-25-36529R1

PLOS ONE

Dear Dr. Osei Duah Junior,

I'm pleased to inform you that your manuscript has been deemed suitable for publication in PLOS ONE. Congratulations! Your manuscript is now being handed over to our production team.

Kind regards,

on behalf of

Dr. Javier Fagundo-Rivera

Academic Editor

PLOS ONE